# Restoring the Importance of Cereal-Grain Legume Mixtures in Low-Input Farming Systems

Jerzy Księżak [1], Mariola Staniak [1] and Jarosław Stalenga [2,*]

1   Department of Forage Crop Production, Institute of Soil Science and Plant Cultivation-State Research Institute, Czartoryskich 8, 24-100 Puławy, Poland
2   Department of Systems and Economics Crop Production, Institute of Soil Science and Plant Cultivation-State Research Institute, Czartoryskich 8, 24-100 Puławy, Poland
*   Correspondence: stalenga@iung.pulawy.pl; Tel.: +48-81-4786-808

**Abstract:** In the paper, we overview the benefits of cereal-grain legume mixtures in low-input farming systems and identify the key factors affecting their agricultural performance. The review was based on the data collected from databases such as Google Scholar, Web of Science, and ScienceDirect. The gathered literature covered the original research articles, reviews, book chapters, and, in a few cases, conference proceedings. The focus was on literature coming from Mediterranean countries and Central-Eastern Europe, especially from Poland. Originating from two different plant families, legumes and cereals complement each other. The legumes provide biologically fixed nitrogen for the cereals in the mixture, but also increase its pool available to the following crops. Additional benefits include, but are not limited to more efficient pathogen and weed control, supply of high-quality fodder, and improved economic efficiency. Cultivation of crops after such mixtures usually results in higher and more stable yields. The productivity of cereal-grain legume mixtures largely depends on the soil factors (soil type, pH, water availability, etc.), crop species, crop variety, and crop management. Cereal-grain legume mixtures are particularly relevant to the poor (sandy) soils which are often unsuitable for the production of the components grown as a sole crop and are often linked with low-input farming systems.

**Keywords:** cereal-grain legume mixtures; nitrogen fixation; intercropping

## 1. Introduction

Crop mixtures or intercrops are defined as two or more species grown together at one time. They date back to at least 300 B.C. [1] and have been commonly used worldwide. It is assumed that even at the end of the 20th century, the traditional, multiple and mixed cropping systems located mainly in the areas with less intensive agriculture, typical for Africa and Asia, provided as much as 15–20% of the world's food supply [2]. In Europe, intercrops have significantly reduced their area through the 20th century with an increase in mechanization and chemical intensification of agricultural production. Today the intercrops are most commonly found in low-input and organic farming systems [3]. Successful mixtures consist of the crops having complementary rather than competing traits and thus using resources more efficiently than the sole crops [4,5].

In recent years, the concept of ecological intensification has been developed [6]. This concept aims to increase the contribution of natural cycles and resource flow in agricultural production by wide utilization of supporting and regulating ecosystem services provided mainly by beneficial organisms [7,8]. Cultivation of crop mixtures, especially with legume components, seems to be a worthy supporting option for this intensification.

Cereal-grain legume mixtures (CGLM) may be grown for grain feed, green forage, silage, or for green manure, etc.; intercropping has been shown to over-yield [5,9], boost the forage protein in rations [10], and stabilize crop yields [11]. The yield advantage of mixed

stand in relation to sole crops mostly lies in more efficient utilization of light, water, and/or nutrients by a complementary foraging pattern of the associated species, which implies lower interspecific than intraspecific competition for these resources [12,13].

The benefits of CGLM cultivation include less severe weed infestation [14,15] and higher resistance to pathogens, such as those causing wheat flag leaf diseases [16]. Moreover, the exchange of nutrients, nitrogen in particular, between such species (inter-plant exchange transfer) can highly contribute towards low external nutrient management. Recent rises in fertilizer costs, coupled with concern about the environmental impacts of excess nitrogen use and associated legislation are driving farmers to seek alternative nitrogen sources and approaches such as intercropping [17]. Additional benefits include improved quality of the fodder and high pre-crop value [10].

Most of the existing reviews on the performance and benefits of cereal-grain legume mixtures concern conventional, usually high-input, farming systems located mainly in Western European countries and usually take into account broadly defined intercropping [1,3,5,10,18–20]. Here we have concentrated on the results of research conducted *mainly* in low-input farming systems, located primarily in Mediterranean countries and Central-Eastern Europe. Restoring the importance of cereal-grain legume mixtures in low-input farming systems is particularly important in the context of the overwhelming dominance of cereals in the cropping pattern in such systems. For example, in 2019, in Poland and in the Baltic states (Latvia, Lituania, and Estonia), the share of cereals in this pattern amounted to 75–80% [21].

The aim of this paper is to identify the key factors affecting the agricultural performance of cereal-grain legume mixtures and to overview their main benefits mostly from the perspective of the low-input farming systems.

The review was based on the data collected from different databases, mainly Google Scholar, Web of Science, and ScienceDirect. The scope of this review was limited to the terms: "cereal grain legume mixtures", "intercrops\intercropping", "ecosystems", "organic systems", and "legume nodule", which were searched in the article title, abstract, keywords, and subject within each database. The reviewed literature covered the original research articles, but also reviews, book chapters, and, only in a very few cases, conference proceedings. Grey literature (e.g. technical reports, books, Ph.D. thesis, and Master's dissertations) was excluded.

## 2. Key Factors Affecting the Performance of Cereal-Grain Legume Mixtures

### 2.1. Soil Type

The yield of CGLM largely depends on the soil type, soil pH, and water availability, which is important under conditions of high variability of climatic conditions. Mixed crops are characterised by greater yield stability compared to monocropping, especially on poor, sandy soils. According to Księżak and Magnuszewska [22] the largest yield of pea and cereal mixtures was noted on soils with neutral pH, and the smallest, on very acidic. Rudnicki and Kotwica [23] pointed out, that on medium-textured soils, the yield of cereal-lupine mixtures depended mainly on the cereal component. Species of lupine (yellow (*Lupinus luteus* L.), blue (*Lupinus angustifolius* L.), and its sowing density had no significant impact on the overall yield of the mixture. However, on the heavier textured soils, most suited to wheat (*Triticum aestivum* L.) production, CGLM differed very much in height, yield, the share of a legume component, and other characteristics [23].

Another important factor influencing the yield of CGLM is the availability of soil water. At low soil moisture content, the mixture of peas (*Pisum sativum* subsp. *arvense* (L.) Asch.) with cereals yielded better than pure sowing of these species [24]. According to Dudek and Żarski [25], the use of irrigation allowed for the cultivation of mixtures for green fodder even on very light (sandy) soils, providing higher (by 163%) and more reliable yields.

### 2.2. Choice of Species

Legume-based mixtures show positive interspecies interactions that justify their association with other species. CGLM perform well under temperate climate conditions and yield better and more stably than the same species grown in pure sowing. When composing a balanced mixture, attention should be paid to the duration of the growing season of the CGLM components and matching the harvesting time, as these are important factors affecting their yields. It is also related to the competitiveness of plants. Most cool-season annual legumes suffer from a competitive disadvantage in association with cereals. This serious disadvantage has usually been reported for peas, white lupine, and vetches [26]. According to Księżak and Borowiecki [27] peas mixed with wheat yielded better than with barley (*Hordeum vulgare* L.), while oats (*Avena sativa*) as a companion crop for lupine gave smaller yields than triticale (× Triticosecale) [28]. Yields of lupine with oats were small and variable between years, and the share of lupine seeds in the yield of these mixtures generally did not exceed 10%. On poor, sandy soils, yellow lupine in mixtures with spring cereals performed better than blue lupine [23]. Droushiotis [29] showed that the higher share of legume (vetch or pea) seeds in the mixture with triticale or barley, the smaller its total dry matter production. According to this author, the highest proportion of legume was noted in the mixture of pea with triticale, assuming the harvest time for the cereal was the phase of grain milk maturity, and for the pea the final stage of the pod formation.

Herper [quoting after 5] argues that if the components of the mixture compete with each other in terms of height, the overall yield is determined by the weaker component of the mixture. Baxevanos et al. [30] suggest that the ideal pea plant type depends on the extent of competitive stress as determined by the intrinsic plant vigour and plant height of the associated cereal and the presence and amount of nitrogen fertilization. A tall semi-dwarf type is preferable under modest competitive stress, while a tall type is preferable under severe competitive stress. Plant modelling work by Barillot et al. [31] on pea-cereal mixtures suggested that pea competitive ability was positively affected by leaf area index (LAI) in early growth stages, and by plant height during the onset of interspecific competition.

Horse bean (*Vicia faba* L.) is one of the more popular grain legumes and appears to be very suitable for intercropping with cereals. It has good spatial and temporal complementarity with cereals in most climates [21] and presents less risk of yield failure than sole crops [1] standing. Its global production stands at 4 Mt (from 2.6 M ha) [21]. Horse beans after inoculation with Rhizobium strains can fix up to 88% of their nitrogen requirements and leave considerable quantities of nitrogen in the soil after harvest [32].

Little attention has been paid to root system effects. Wilson [33] argues that overyielding of mixtures (i.e., when the yield of the mixture is higher than the yield of the components treated separately) is associated with diversity in the root system structure of its components. Księżak [34] showed that the substrate in which seeds of four species of cereals germinated (wheat, barley, oats, and triticale) stimulated the growth of rootlets of pea and vetch (*Vicia sativa* L.). The corresponding substrate from triticale inhibited the growth of yellow and blue lupine rootlets. Moreover, the substrate from beneath the seeds of legumes had an inhibitory effect on the development of rootlets of cereals, as well as on the coleoptile of barley. Księżak and Staniak [35] reported that root secretions from oat seedlings inhibited the development of the rootlet of legumes after 96 h. The grain extracts from oats stimulated the germination of pea seeds after just 24 h, but no such effects were observed in the case of barley exudates. Secretions from pea seeds soaked for 48 h had an inhibitory effect and after 72 h strongly inhibited the germination of barley [35].

### 2.3. Choice of Varieties

The positive effect of interspecific diversity with legumes on crop yielding is widely acknowledged. In contrast, the effect of large intraspecific diversity within each associated species is little studied. Greater intraspecific diversity of annual legumes, as provided by genotype or variety mixture, reveals some advantages in terms of yield and stability [12]. Among different traits, plant height is of particular importance, as it strongly determines

the architecture of the canopy, and affects lodging and yields of mixtures. Rudnicki [36,37] developed a methodology for assessing the suitability of varieties of pea, yellow and blue lupine for mixtures with spring cereals, at the same time providing formulas for determining the composition of such mixtures. The author took into account such aspects as time of maturity, plant height, resistance to lodging, thousand seeds weight, and the protein content in the seeds. According to this author, the height of plants in the mixture and the habitat type determine the stand architecture. The varieties with a shorter stem usually have less favourable light conditions in the mixture. Particularly unfavourable conditions occur when legumes dominate over cereals, which may lead to the lodging and, consequently, to yield decrease [38]. Well-chosen varieties of pea increase the chance of over-yielding in mixtures with barley, triticale, and oats [38]. Semi-leafless pea varieties have a high-yielding potential and a lower rate of transpiration due to a large number of tendrils. Tolerance of varieties to the neighbourhood of other crops is also important, as is resistance to pod shatter In legumes. Nykänen et al. [39] showed that for tall varieties of forage pea, oats, and wheat were better components than barley. Księżak [24] indicated that determinate varieties of vetch yielded better with spring barley than with oats. Darras et al. [40] showed that pea genotypes cultivated as isolated cultivars in mixtures with barley demonstrated the potential to improve field pea competitive ability, particularly with respect to the least competitive genotypes, but this result emerged mainly in suboptimal growing conditions.

*2.4. Seeding Rate*

The yield of a CGLM depends on the performance of its weaker component. A low proportion of legumes in the mixture reduces the proportion of their seeds in the yield, but also makes these plants more vulnerable to yield fluctuations. In addition, it reduces biological nitrogen fixation and compromises yield quality. Therefore, it is important to optimize the proportion of components in the seed mixture, especially the legume one, taking into account the species, habitat, and crop management. There are uncertainties regarding the effect of sowing density on the grain yield of CGLM. Particular species in the mixture can be sown at either their monocrop seed rates (an additive design) or at a reduced percentage (a replacement design). Most research has been based on the replacement design of 50:50 legume:cereal [15,41,42], but there are designs such as 60:40, 80:20 [13], 63:37, 75:25, 85:15 [43], 65:35, 55:45 [44], along with additive designs of 100:100, 50:50, and 100:50 [15,41]. Haymes and Lee [45] have shown up to a 40% grain yield increase for intercrops compared with sole crops (Land Equivalent Ratios (LER) up to 1.4). The 100:100 designs have shown a constraint to the legume due to competition from the cereal, but a general intercrop advantage has been widely reported. The other designs indicated the advantages of the legume:cereal crop combinations on weed suppression, protein yield, and nitrogen uptake. In these studies, two factors were taken into account: MAI (Monetary Advantage Index) and IA (Intercropping Index). In any case, the mixtures were more favourable than objects with plants grown as a sole crop. The proportion of components had a significant effect on the yield of mixtures. The yield of the mixture usually decreases with an increasing share of legumes in the mixture [23,38,46]. This relationship was observed for mixtures of cereals with peas [38,46], as well as for mixtures of cereals with yellow and blue lupine [23].

The results of the meta-analysis of the crop yields in cereal/legume mixtures done by Yu at al. [18] showed that the higher the sowing density of cereal, the higher its relative yield and the higher the yield of the whole mixture. Intercropping of common vetch with different cereals (wheat, triticale, barley, and oat) at two different seeding ratios (55:45 and 65:35) under Mediterranean climate conditions affected the crop yields, competition between cereal and legume and the economic efficiency of the cropping system. The common vetch–wheat mixture at the 55:45 seeding ratio and the common vetch–oat mixture at the 65:35 seeding ratio yielded the best and were the most profitable compared with other intercropping systems [44].

## 3. Key Benefits of Cereal-Grain Legume Mixtures

### 3.1. High Pre-Crop Value

Cultivation of CGLM may contribute to increasing the yields of the following crops [47–49]. This is mainly due to the high pre-crop value of the mixture residues, which deliver large amounts of essential macronutrients to the soil, especially nitrogen fixed by the legume crop. Pappa et al. [47] reported an increase in the yield of oats following an intercrop (barley/pea and barley/white clover (*Trifolium repens* L.) compared with those following a solo crop. CGLM is a good pre-crop for root crops, but especially for winter cereals. Rudnicki and Kotwica [48] showed that the yields of winter wheat cultivated after mixtures of peas with spring cereals (wheat, barley) were larger by 5–27% than after spring barley. In turn, the yield of winter wheat after the mixture of yellow lupine with triticale was higher by 31% than after triticale grown as a sole crop. Rudnicki et al. [49] showed a high pre-crop value of mixtures of oat with legumes (pea or lupine) for winter wheat (Table 1). The results of Monti et al. [50] showed a durum wheat grain yield increase after pea-barley intercrop. Interestingly, this study suggests that despite the fact that crop residues were removed at harvest, the stubble of previous legume crops still explained a relevant part of the variation in subsequent wheat grain yield, since roots and rhizodeposits were rich in nitrogen. Siuta et al. [51] found that the species of cereal in the mixture with peas did not affect the winter wheat yield, however, a higher share of peas in the mixture increased the yield of wheat, especially on better soils. Similar tendencies were revealed for the pre-crop value of mixtures of triticale with yellow lupine [48]. However, Rudnicki et al. [49] showed that for winter triticale grown after mixtures of oats with yellow lupine, the pre-crop effect was weak. The authors pointed out that this was due to the properties of oats, which are treated as one of the best pre-crops for winter cereals. Yields of winter wheat after barley or oats with peas compared to the cultivation after triticale showed very little variation in years [48]. According to Kotecki et al. [52], crop residues of a mixture of lupine with triticale (straw, stubble, and roots) enriched the soil with 32 kg of nitrogen and 55 kg of potassium, while for comparison, horse bean cultivated as a sole crop left in the soil approximately (kg ha$^{-1}$): N 75, P 5–8, K 90–140, and yellow lupine (kg ha$^{-1}$): N 65–75, P 15–18, K 95–120.

**Table 1.** Yields of winter wheat depending on the pre-crop type [49].

| Pre-Crop | Wheat Yield (t ha$^{-1}$) | Wheat Yield (100%) | Variability of Yields (%) |
|---|---|---|---|
| Yellow lupine | 5.28 | 100 | 3.5 |
| Peas + lupine | 5.17 | 98 | 3.8 |
| Oat + lupine | 4.98 | 94 | 1.8 |
| Oat + peas | 4.94 | 94 | 3.6 |
| Oat | 4.79 | 91 | 4.5 |
| Barley + oat | 3.63 | 69 | 13.6 |
| Barley + peas | 3.47 | 66 | 16.7 |
| Triticale + lupine | 3.34 | 63 | 17.3 |
| Barley + lupine | 3.28 | 62 | 20.8 |
| Oat + triticale | 3.23 | 61 | 17.4 |
| Barley + triticale | 3.17 | 60 | 15.6 |

### 3.2. Increased Nitrogen Use Efficiency

One of the ecosystem services provided by CGLM is the biological fixation of nitrogen and the resulting reduction of energy consumption and greenhouse gas emission, protection, and improvement of soil fertility and its physical, chemical, and biological properties and rotational break crop effects. The nitrogen fixed by the legume is utilised by it but is also available to the cereal. It is particularly evident under conditions of limited soil nitrogen availability, where intercropping improves not only the yield level of the mixture but also increases the protein content of cereal grain. Biological nitrogen fixation by legumes living in symbiosis with rhizobia is of great significance for agriculture. In this process, legumes provide the bacteria with carbohydrates, and in return, they receive nitrogen assimilated by

them, which they use to produce high-value protein. In crop rotations, the $N_2$-fixing ability of grain legumes is the main factor for nitrogen contribution in succeeding non-legumes and it can reduce the mineral nitrogen fertilizer demand of the cropping system [50]. The increased nitrogen use efficiency in intercropping of grain legumes and cereals has the potential to globally reduce the requirements for synthetic nitrogen fertilizers by about 26% [19].

Non-legume plants growing in the vicinity of legumes benefit from the nitrogen fixed by legume root nodule bacteria which is transferred to the soil in the form of aspartic acid or β-alanine [41,42,53]. This is especially true under conditions of low soil nitrogen availability, where intercropping improves not only the crop yield but also the cereal protein content and the phosphorus availability for the cereal [12,54]. This phenomenon is especially important in organic farming systems, where biological fixation is the key source of nitrogen. Triboi [55] in lysimetric studies using 15 N showed that vetch sown with oats fixed about 53 kg nitrogen per ha i.e., 90% of the total nitrogen taken up by the stand. The oats used about 28 kg N, i.e., 1/3 of the nitrogen taken up. According to Wysokiński and Kuziemska [56] about 10% of nitrogen taken up by spring triticale cultivated in a mixture with yellow lupine originated from biological fixation by the legume. Fujita et al. [57] pointed out that in the mixture of soybean (*Glycine max*) with sorghum (*Sorghum bicolor* (L.) *Moench*), the transfer of nitrogen from a legume to cereal increased the yield and efficiency of nitrogen utilization in the mixture. Wacquant et al. [58], on the basis of the studies on three mixtures: oats with vetch, Italian ryegrass (*Lolium multiflorum*) with red clover (*Trifolium pratense* L.), and maize with soybeans, showed that secretions from active root nodules of legumes contained ions of $NO^{3-}$, which affected the increase of the biomass and nitrogen content of non-legume components of these mixtures. The meta-analysis (29 field-scale studies) used by Rodriguez et al. [20] confirms and highlights that intercropping consistently stimulates complementary nitrogen use between legumes and cereals by increasing $N_2$ fixation by grain legumes and increasing soil nitrogen acquisition in cereals. Cereals are stronger competitors for inorganic nitrogen thus, forcing legumes to depend to a higher degree on symbiotic $N_2$ fixation.

The complex process of legume BNF is affected by temperature, soil water content, nitrogen concentration, root zone pH, and genetic variation in potential nitrogen fixation capacity. It is also affected by plant nutritional status such as P and K levels that control nodule growth and nitrogenase activity [59]. Generally, soil temperature controls legume BNF (both very high and low temperature reduces nitrogen fixation), however, the response of nodule establishment to soil temperature differs between species (Table 2). Drought stress generally inhibits nitrogen fixation, and the inhibition is reinforced as drought stress becomes more intense [60]. Additionally, soil mineral nitrogen in the root zone inhibits legume nodulation and nitrogenase activity [61].

**Table 2.** Response of BNF of selected legumes to soil temperature (in °C).

| Species | Minimum | Optimum | Maximum | Reference |
|---|---|---|---|---|
| Common bean | | | | Michiels et al. [62] |
| Common vetch | 2 | 20 | | Dart and Day [63] |
| Horse bean | 5 | 20 | 40 | Waughman [64] |
| Field pea | 0.5 | 25 | 40 | Waughman [64] |
| Soybean | | 20–25 | | Lindemann and Ham [65] |

### 3.3. Lower Mineral Nitrogen Leaching

Intercropping cereals with legumes usually allows for a decrease in the use of mineral fertilizers mainly due to crop complementarity and the ability of legume components to fix the nitrogen [66]. The mixture consisting of species that root at different depths, intercepts nitrogen from the soil, reducing its leaching into the surface and groundwater. Hauggaard-Nielsen et al. [67] based on the results from a 2-year lysimeter experiment on a temperate sandy loam soil showed that $NO^{3-}$ leaching tended to be smaller in objects

originally cropped with the pea-barley intercrops, although not significantly different from the sole cropped pea and barley. Pappa et. al. [68] showed that the nitrate leaching in the two field experiments on sandy loam soils in Scotland was reduced in spring barley/pea intercrops when compared to the barley monocrop, however, strong varietal effect for pea on nitrate leaching was revealed. Mariotti et al. [69] concluded that the intercrop of field bean with spring barley was characterized by lower $NO^{3-}$ leaching than field bean as a sole crop, however, it was higher than the barley as a sole crop.

### 3.4. Improved Efficiency of Pathogen Control

Intercropping can be also an efficient method to control pathogens [19]. Legumes intercropped with cereals have a lower incidence of fungal diseases than those grown in pure sowing. Cereals also act as a barrier to some pests, limiting their spread. Fernandez-Aparicio et al. [70] showed that in the Mediterranean climate chocolate spot incited by *Botrytis fabae* was efficiently controlled when horse bean was intercropped with different cereals (barley, oat, triticale, and wheat), but not when intercropped with other legumes. Additionally, for Ascochyta blight mainly caused by *Mycosphaerella pinodes*—one of the most serious pea diseases—the damage was significantly reduced above 60% in pea intercropped both with triticale or horse bean compared with pea growth as a sole crop [71]. According to the authors in both cases, suppressive effects may have been a consequence of a combination of host biomass reduction, altered microclimate, and physical barriers to spore dispersal. Ndzana et al. [72] showed that a mixture of winter pea with durum wheat significantly decreased the pea aphid (*Acyrthosiphon pisum* Harris) abundance in comparison to the sole crops.

### 3.5. Better Control of Weed Infestation

Control of weed infestation in sustainable farming systems involves the use of direct methods, involving interventions into the stand and indirect methods of preventive character, such as proper crop rotation, choice of varieties with greater competitiveness against weeds, proper agronomical practices, and the use of undersown crops and mixtures [73–76]. CGLM competes with weeds stronger than sole crops, but it is also dependent on the composition of the mixture, the share of components, as well as soil conditions [77,78]. In general, an increase in the proportion of legumes in the canopy results in less competitiveness against weeds. In pea-barley intercrops, the presence of barley supported the growth of peas and the establishment of a dense canopy with this resulting in more effective weed suppression than horse bean based [50] or the peas sole crops [15]. The study of Staniak and Księżak [79] showed that among four different mixtures of oats with peas, oats with vetch, barley with peas, and barley with vetch, with 50% share of the components at sowing, the mixture of barley with peas was the weediest, as evidenced by the largest biomass and number of weeds. Of all objects, the mixture of oats with vetch was the most competitive to weeds. The findings of Šarūnaitė et al. [77] indicated that among the four mixtures of spring wheat with grain legumes, such as peas, lupine, vetch, and horse bean, the mixture of wheat with vetch limited weed infestation the most, while the least competitive was the mixture with lupine. The highest weed infestation was recorded in the sole lupine and pea. In general, the higher share of grain legumes in the mixture the higher weed infestation, which indicates higher competitiveness of cereals than grain legumes in relation to weeds. The favourable effect of mixtures on reducing weed infestation reveals more in wet years [79,80].

### 3.6. Source of High Quality Fodder

Cereal-grain legume mixtures are mainly cultivated for green forage, silage, or for grain. Intercropping of these species provides a high-quality forage that has a high protein content with a better amino acid profile than cereals grown in mono-crops. This makes the silage efficiency of the mixtures higher than that of cereals, allowing higher animal gains in a shorter period of time. Crops designed for green forage or silage are usually

harvested in the early dough stage of cereals. Silage made from a CGLM gave similar effects as meadow grass in the feeding of calves, although the intake of silage is somewhat lower. Kraszewski et al. [81] found that the use of silage from the legume-cereal mixture (sown in the amount of 50 kg of barley, 50 kg of oats, and 70 kg of field pea) in the feeding of young bulls mixed with maize silage at the ratio of 1:1 gave good weight gain and slaughter results, as well as having positive effects on the quality of meat.

Intercropping triticale or barley with legumes has been commonly used to increase the protein content of low-input silage production systems in dry areas. Legumes cultivated as a sole crop have shown many limits due to low and unstable yields and high susceptibility to disease and pest attacks. Compared with others, pea-triticale and pea-wheat mixtures showed a positive response in terms of productivity and quality of forage, which makes them a good alternative to the sole crop for forage production [1].

Brundage and Klebesadel [82] found that the level of protein in oats intercropped with peas decreased from 20% of the dry matter before earing to 10% in the maturity phase, while in peas, it remained at the same level (about 15%) from the beginning of intensive growth. The content of crude fibre and an acid detergent fraction (ADF) was higher in oats than in peas, and lignin content in peas was steady at the level of 5% dry matter, whereas in oats, increased from the milk phase to full grain maturity by approximately 5%. Taking into account the yield and digestibility of dry matter, the authors indicated the period from the milk stage to the early dough stage of oats as the most appropriate time for the harvest of the mixture. Blade et al. [83] in their study demonstrated that a mixture of peas with barley allowed obtaining better quality silage compared to mixtures with triticale. A higher level of protein and lower neutral detergent fibre (NDF) content clearly showed the advantage of using such a mixture.

The energy value of dry matter expressed in feed units for lactation (UFL) and feed unit for maintenance and meat production (UFV) was more favourable in the case of a mixture of peas with barley than with oats. However, a mixture of peas with oats was better in terms of the nutritional value of protein [84]. Wawrzyńczak et al. [85] showed that silage from the whole-crop biomass of legume-cereal mixtures allowed for achieving large weight gains of young bulls and that the nutrients from such a feed were efficiently used. In other research, Pisulewska [46] noted that a mixture of spring triticale with field pea was characterized by the highest nutritional value of protein (EAAI) and high content of lysine, isoleucine, and threonine. Pozdisek et al. [86] showed the better nutritional value of organic feed for monogastric animals originating from mixtures of peas with spring wheat and pea with spring barley in comparison to crops cultivated as sole. Moreover higher crude protein content of barley grown in a mixture with peas was noted. This increase was, however, compensated by a decline in NFE content (nitrogen-free extract). Additionally, wheat and barley grown in a mixture with peas contained more methionine than cultivated as a sole crop and barley contained more threonine.

In organic farming, the cultivation of CGLM significantly increased protein concentration in fodder [87,88]. Lithourgidis et al. [89] noticed that the vetch in the mixture with oats or triticale favourably affected the quality of feed. The high content of protein was also recorded in the mixtures of peas with oats, but in terms of the feed quality, mixtures of peas with barley or triticale were favourable as well [23]. Similar results with pea/barley mixture to over-yield the sole crops have been shown in northern countries [4].

### 3.7. Improved Economic Efficiency

Growing mixtures of cereals with grain legumes is more economically efficient compared with pure legume sowings, due to higher productivity, yield stability, and lower use of pesticides and mineral fertilizers. Pelzer et al. [66] showed higher average gross margins for pea-wheat intercropping compared to sole pea and sole wheat cultivation in unfertilized conditions and slightly lower gross margins under fertilized conditions. The authors also found, that pea-wheat intercropping led to higher profitability if it replaced pea or if both crops were cultivated and replaced by the intercropping system e.g., 2 ha

pea-wheat was more profitable than 1 ha pea and 1 ha wheat. In contrast, Hauggaard-Nielsen and Jensen [4] reported higher economic benefits of pea sole cropping compared to pea-barley intercropping and barley sole cropping. However, the yields of pea (6 Mg ha$^{-1}$) in the experiment were above the average for the studied region (4.5 Mg ha$^{-1}$). Pea-barley intercropping resulted in a higher net income compared to barley sole cropping, especially in unfertilized systems.

Prices for grain produced in intercropping systems are generally equal to grain produced in sole cropping except if the grain has a higher protein concentration. It is therefore assumed, that increased protein content of wheat grain in intercropping could be of economic benefit when selling it for bread making [90]. However, this is only the case if the legume crop can also be marketed effectively. Intercropping systems have also the potential to reduce the application of inputs i.e., nitrogen fertilizers and agrochemicals, due to the nitrogen-fixation of legumes and their ability to suppress pathogens and weeds [14,65,91].

When analyzing the economic benefits of intercropping, except for the productivity, quality, and variable costs for machinery use and inputs, the costs for grain sorting need to be also considered [4,65].

### 4. Conclusions

Intercropping of cereals and grain legumes can be an important measure for increasing the efficiency and resilience of modern agriculture, stabilising crop productivity while providing many ecosystem services. Different farming systems, but especially low-input and organic, may benefit from the use of cereal-grain legume mixtures. Key benefits include increased input of nitrogen from biological fixation rather than from fertilizers and consequently high pre-crop value for the following crops, lower mineral nitrogen leaching, improved efficiency of pathogen and weed control, production of a high-quality fodder, and in many situations, better economic efficiency.

The results presented in this review reveal intercropping of cereals with grain legumes as a valuable option in rotations with cereals in dry areas and on poor (sandy) soils managed with low inputs, where the net income of the legume system is not comparable to cereal crops. Such a set of conditions is quite common in agriculture of typical for Central and Eastern Europe.

There is a need for further research on other legume species, including those that have been grown in a small area under temperate climate conditions (e.g., soybean). There is also a research gap in the area of relationships and interactions between different mixture traits (e.g., root complementarity), as well as greater intraspecific diversity in legumes.

**Author Contributions:** Writing—original draft preparation, J.K.; writing—review and editing, M.S. and J.S.; visualization, J.S. and M.S. All authors have read and agreed to the published version of the manuscript.

**Funding:** This research received no external funding.

**Institutional Review Board Statement:** Not applicable.

**Conflicts of Interest:** The authors declare no conflict of interest. The funders had no role in the design of the study; in the collection, analyses, or interpretation of data; in the writing of the manuscript, or in the decision to publish the results.

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
