# Peer review of "Restoring the Importance of Cereal-Grain Legume Mixtures in Low-Input Farming Systems"

_agriculture, doi:10.3390/agriculture13020341_

Round 1

Reviewer 1 Report (New Reviewer)

Thank you for the opportunity to review this manuscript. In my opinion, the content is suitable for publication in MDPI Agriculture, although, some revision is necessary for clarification and the focus should be globalized, which will require additional literature review not limited to geographic region. There is a high rate of self citation; however, because of the limited region of the database to review, that would be expected, but globalizing the review will likely reduce the self-citation rate as considerable work has been done on the same topic elsewhere.

Author Response

Thank you very much for your valuable comment. This is a very important point you touched. When planning the manuscript we noticed that there are already published review papers looking at the cereal/legumes mixtures from a global perspective. We found however a research gap in the area on how this type of intercropping would perform in extensive farming systems of Central/Eastern Europe where specific climate and soil conditions (mainly continental climate and poor/sandy soils) significantly reduce the possible scope of suitable crops. This is the reason why the authors also used their own publications to some extent. Our review attempts to make a comprehensive synthesis on the basis of these and many other papers. 

Reviewer 2 Report (New Reviewer)

Comments to Editor:

The authors investigated the “Restoring the Importance of Cereal-Grain Legume Mixtures in   Extensive Farming Systems of the Central and Eastern Europe.”

·         While I consider that the study and findings are potentially interesting, I have minor concerns that must be addressed in revising this manuscript.

·         The authors have not explained the research problem and research question clearly.

Based on my comments below, I recommend a minor revision of the manuscript.

Specific Comments

Abstract

The abstract needs to be revised (e. g., give details of the main results and link your results systematically and name the methods to achieve the goal/aim described in the background).

Introduction

The introduction does not provide enough information on the research background and gap. Therefore, it is strongly suggested that the authors please add more relevant literature on their research question.

Results

Put an introductory/general statement about your main finding of each result as the first sentence.

Discussion

Authors should rethink what they write in the first paragraph and only summarize the main findings given the research questions.

Conclusion

It is strongly suggested to authors that please revise their conclusion and just give recommendations for future studies with shortcoming of your study. 

Author Response

Thank you very much for your valuable comments. We have made changes in the manuscript in accordance with your suggestions.

The Abstract was revised, the key results and the methodical approach was added to it.

In the Introduction the research gap was presented, i.e. how intercropping of grain cereals and legumes would perform in extensive farming systems of Central/Eastern Europe where specific climate and soil conditions (mainly continental climate and poor/sandy soils) significantly reduce the possible scope of suitable crops. There is some review literature on this but only from a global perspective or from other European regions: [1, 3, 5, 10, 44, 51, 58]. These publications were cited in the Introduction.

In the Results/Discussion a general/concluding statement about main findings was added at the beginning of each part.

The Conclusion was significantly revised and  recommendations for future studies were added.

The English will be improved by editing service of MDPI.

Round 2

Reviewer 1 Report (New Reviewer)

Comments to Authors: MDPI Agriculture- 2054599, “Restoring the importance of cereal-legume mixtures in extensive farming systems of central and eastern Europe”

I appreciate the authors' diligence in revising the manuscript. Thank you!

I only found a few things that need to be addressed:

Line 60 and elsewhere: I am still concerned about the clarity of "pre-crop" and "forecrop". It seems that you are referring to the benefit of mixtures to the following crop, as shown in the revision at line 445. Consequently, I recommend using text similar to that when referring to the CLM in the text and using ". . . depending on the preceding crop type in Poland, . . ." in the Table 1 title. For references, I think it fine to still use 'pre-crop' in the translated titles. 

Lines 306 & 344: While, it was highlighted, there was no change in the text at line 306 for 'object' it seems that at line 344, you replaced 'objects' with 'treatments' but the pdf of the R4 version seems to indicate that you inserted 'treatments' and then deleted it instead of 'objects.'

Line 385: It seems that 'obtaining' was meant, but it reverted to 'obtained'.

Line 431: My apologies, I left the 's' off 'systems'. It should read 'systems also have the'.

References: Check the date format for references 8, 23, 35, 47, & 83. Recently published articles in MDPI Agriculture, including those dates inserted by the editors showing publication progress, follow a day month year format, with the month spelled out rather than DD.MM.YYYY. For example, check the publication progress dates (received, revised, accepted, and published) and date in reference #2 in  https://doi.org/10.3390/agriculture12010115.

Thank you for your diligence in revising the manuscript so the information you have gathered and presented very well can be available tothe scientific and producer communities of interest!

Author Response

This manuscript is a resubmission of an earlier submission. The following is a list of the peer review reports and author responses from that submission.

Round 1

Reviewer 1 Report

The title is interesting the authors provide an over-view of the benefits of CLM and identify key factors affecting their agricultural performance. Is it confined to Europe?

There are 84 references included whereof 8 (10 %) are conference proceedings. A few are articles from regional journals. The most recent article included is published in 2018, and many are dated back to the 1990´s. The recently published articles are missing. There are publications by Jensen, E.S. form 2020 and by Calsson, G. 2021, to mention some.

Advise: Find the literature published from 2015 and onwards. Exclude conference proceedings and include peer reviewed articles.

Language: You use pure stand and sole crop irregularly. I prefer sole crop.

The most serious remark is that you use the references in a wrong way. You are supposed to “tell a story” as in lines 251 -256. You must line up the information and put the author at the end of the sentence as digits within brackets. This is the main reason to my decision to reject the article along with the incomplete review of recent publications.

Some advice for improvement of a future submission:

Line 15 Originating will replace coming from

Line 18 You don´t give any examples of pest control

Line 19 What are your definitions of poor soils?

Line 21 change pure stand to sole crop

Line 44  Here examples of pathogens and pests  will be of interest.

Line 66 Include latin names for yellow lupin and blue lupin

Line 71 “Under its low levels”? Include hydrotechnical “language”

Line 79 and onwards: you mix pea and cereal , or when a was mixed with b

Line 79 definition of good soils

Line 88 high cereal? Tall spieces or cultivars?

Line 99 Inoculation with?

Line 109 unclear about the 24 hours

Line 123 of the crop, a yield increase

Line 152 Another study?

154 yiel level

Line 176 This section really need clarifying

Table 1 Heading. The results are from Polish studied. Please indicate that in the heading.

Line 200 Leguminous plants enter symbiosis with nitrogen fixing bacteria called rhizobia.

Rhizobium is a genus, and you have to explain it.

Line 204 The evidence?

Table 2 Optimum range describes an interval. You mean optimal temperature? Include degree C.

Line 238 and onwards no pest control is mentioned.

Lines 251-255 Here you tell a story!

Line 274 When you start a new section it is easier for the reader to follow if you write Cereal mixed legumes

Lines 289-299 a section with a story!

Reviewer 2 Report

Specific comments

As a researcher in the field of farming, I am very interested in your work. I have looked thoroughly at your article and I see that you did a lot of work on it.

However, There are some problems in the article that need to be solved, if I understand your description correctly. As far as I see, the paper can be accepted if the points below are dealt with appropriately.

Topic

1. Specifically differentiate between “mixtures” and “mixed cropping”.

Abstract

2. “CLM are mainly grown for grain feed, green forage, silage or for green manure.” This sentence needs to be written in the introduction, not in the abstract.

3. “Cereals growing in the vicinity of legumes benefit from nitrogen assimilated by legume root nodule bacteria.” This sentence is suggested to be deleted.

Interduction

4. “Cereal-legumes mixtures (CLM) may be grown for grain feed, green forage, silage or for green manure.” There is no literature cited in this sentence.

5Introduction- Line 30-32 “In Europe, intercrops largely disappeared through the 20th century with an increase in mechanization and chemical intensification of agricultural production” consider whether the disappearance of intercropping crops is accurate?

6Line 36 “In recent years, the concept of ecological intensification has been developed” Please explain “ecological intensification”

Key factors affecting the performance of CLM

7. “Księżak and Magnuszewska[17] reported the largest yields of mixtures on soils with neutral pH, and the smallest, on very acidic.” Write a single author when quoting from the paper. The full text to modify.

8. In 2.1, suggestions add the effect of CLM on saline-alkali soil.

9. In 2.2, “mixtures of pea with wheat yielded better than with barley on good soils.” This sentence is suggested to be amended to “Peas are mixed cropping with wheat yielded better than with barley on good soils.”

10. “Rudnicki [29,30] developed a methodology for assessing the suitability of varieties of pea and yellow and blue lupine for mixtures with spring cereals, at the same time providing formulas for determining the composition of such mixtures.” and “The author took into account such aspects as: time of maturity, plant height, resistance to lodging, thousand seeds weight, and the protein content in the seeds.” need to be modified. The latter sentence needs to be preceded.

11. Please add references after each design seeding ratio.

12. In 2.4, “The 50:50 designs have generally shown a strong advantage of intercropping across different countries (Germany, Denmark, Italy, UK, Germany and Finland).” is recommended to delete one of them Germany.

13. Please add references after “According to many authors, the yield of mixture decreases together with an increase of the share of legume seeds in the mixture.”

14. Specifically differentiate between “mixed croppng” and “intercropping”

15. In 2.5, please add the nitrogen fixation of legume crops in cereal and legume mixture.

16Line 64 “CLM can often be grown on weaker soils than the sole crops”. How to distinguish weaker soils?

17Line 67-69 “However, on the heavier textured soils, most suited to wheat production, CLM differed very much in height, yield, the share of a legume component and other characteristics” lack of literature.

18Line 85-87 “This author indicated mixture of pea with triticale as the best, assuming that the most appropriate term for the harvest of cereal is the phase of grain milk maturity, and for peas-the final stage of the pod formation.” what is the standard of as the best?

19Line 90-91 “Herper (quoting after 5) argues that if the components of the mixture compete with each other in terms of height, the overall yield is determined by the weaker component of the mixture”. Where is the connection with the previous sentence

20L109-112 “The grain extracts from oats (after 24 hours) stimulated the germination of pea seeds, but no such effects were observed in the case of barley exudates. Secretions from pea seeds soaked for 48 hours had an inhibitory effect and after 72 hours strongly inhibited the germination of barley.” lack of literature.

21Line 155-158 “The common vetch–wheat mixture at the 55:45 seeding ratio and the common vetch–oat mixture at the 65:35 seeding ratio yielded the best and were the most profitable compared with the other intercrop-ping systems.” Whether price factor is considered?

22Line 161-162 In general, yield of mixtures of rye or triticale with vetch increased by about 9.5 kg per 1 kg of nitrogen applied. With the increase of nitrogen, the yield increases. Is it within a certain range.

23. In accordance with lines 57-59, you should either restructure the content below or adjust the content of lines 57-59.

24. Line 87: Please use hyphens correctly.

Key benefits of CLM

25. In 3.1, Lack of summative conclusions. Please add.

26. In 3.2, “the biological fixation is the key source of nitrogen.” Please add references to this sentence.

27. Suggest to delete “In relation to water”.

28. References to pests are missing in 3.4. Please add.

29. Latin name error for triticale in 3.6, please fix it.

30. In 3.6, Lack of summative conclusions. Please add.

31. There are some problems with the data in “yields of pea (6 Mg ha-1) in the experiment were above the average for the studied region (4.5 Mg ha-1).”, please check.

32Line 197 “Yields of winter wheat depending on the pre-crop type” It is better to add significance.

33. Please check the units of wheat yield (100%) in Table 1.

34. In table 2, there are no units in the table header, making it difficult to capture what the key is.

Conclusions

35. CLM also improves soil quality, please add to the conclusion.

References

36. Some years in the references are not bold, please bold.

37. The DOI format in the references is inconsistent, please modify it.

38. There are formatting errors in some references, please correct them.

39. Please mark references DOI in blue.

40. There are many references to the literature, but there is very little of our own thinking in it.

Reviewer 3 Report

Page 1: Line 21 and 20---- avoid sentence beginning with abbreviations

Page1 line 13 avoid using i.e. in abstract

Page 1 line 17 and 18 – explain meaning of ‘high pre crop value for the following crops’

Abstract needs thorough revision and outcome of your meta analysis of the literature 

Page 2 line number 40- No need to abbreviate CLM at many instances

Page 2 subhead 2.1 most of the facts is already known No comprehensive information, it look like review of literature of the thesis dissertation

 Line number 79 scientific names of pea and barley

Line number 80 scientific names of Lupin and triticale

Line number 81 scientific name of oat

Line number 93 and 94- Generalised statement based on single review, its not appropriate

Line number 99 scientific name of faba bean

Line number 105 scientific name of vetch

Subhead 2.5 line 161; across the world lot of research work has been done on N fertilization in cereal-legume mixture

No much detailed reviews on N fertilization in CLM mixture, most of the sentences are case specific

Line No. 174 Remove first sentence,

Line #176 scientific name of clover

Line # 180 remove cropped

Line #194 scientific name of field bean

Line #211 scientific name of soybean

Line #213 scientific name of ryegrass

Line #232- 2 replace with two, numbers below 10 should be expanded in a running sentence (Ex: line 337, 338)

Line # 274 paragraphs should not be start with abbreviation

 Many of the sentences are well known facts, recent developments in cereal-legumes research was not addressed.  Thorough review of the existing literature was not done, just listed as dissertation review of literature. A few legume species were listed and relate with cereals. The review is not comprehensive, not much novelty in the paper